# Resolving Security Issues in the IoT Using Blockchain

**Hafiz Abid Mahmood Malik** [1,*], **Asghar Ali Shah** [2], **Abdul Hafeez Muhammad** [2], **Ahmad Kananah** [1] **and Ayesha Aslam** [2]

1    Faculty of Computer Studies, Arab Open University Bahrain, A'ali 732, Bahrain
2    Department of Computer Science, Bahria University, Lahore 44000, Pakistan
*    Correspondence: hafiz.malik@aou.org.bh

**Abstract:** The Internet of Things is a system of interconnected smart devices that can communicate with each other and with other devices over the Internet, with or without human-to-human or human-to-computer interaction. Although IoT devices, which have IPs, make living easier, they are also a threat to the security and privacy of people. This research work presents a solution to the problem of security in the network of IoT, based on the idea of implementing the blockchain in IoT. Blockchain is a decentralized technology that adds blocks at the end of the chain. It saves the hash value for every block, and corresponds to the previous block. The decentralized behavior of blockchain is best for IoT as an extensive network, because IoT must not have a single point of failure, and one entity must not decide what to do. All the capable storage devices will save the same data entered from any device, removing the risk of receiving altered data.

**Keywords:** IoT; blockchain; cybersecurity; IoT layer; blockchain layer; distributed layer

## 1. Introduction

The current era is one of technology, and a decision's success or failure depends on data. The COVID-19 pandemic has generated a considerable amount of fear globally and has caused a significant number of deaths. Healthcare workers are doing their best to fight against this disease. The IoT plays a vital role in real-time health monitoring, tracking, and reporting of the disease in this situation. In today's world, everything depends on human beings being able to obtain all types of information. Even modern computers and the Internet include the information physically added by humans by typing, pressing buttons, scanning bar codes, capturing pictures using digital or analog cameras, and many other convenient means of adding records. Unfortunately, humans are not good enough to capture accurate data from the real world, which is also a problem. IoT is defined as the network of physical objects that can be located independently and communicate [1]. Kevin Ashton used the phrase "Internet of Things" [2] as the name of his presentation. Since then, the Internet of Things has become a popular topic, from titles of articles to the name of the European Union Conference. To decrease human effort and obtain more information for a better world, IoT has been used in sensors and actuators [3] to collect data and enable them to communicate with each other. However, with increasing ease comes the disadvantages of technology. The main drawback is data breaches. To maintain the privacy of individuals, it is necessary to protect their corresponding data.

The IoT is a resource-constrained field with many challenges; in addition, because it is the beginning of commercial use of this field, much more work needs to be done.

The five key challenges reported by [4] can be stated as:

Limited computation power and memory: Almost all the devices have limited computation capacity due to the limitation of battery life.

Insecure network options: To be easily available to a network, these devices allow insecure and vulnerable networking options such as Bluetooth and do not support complex communication protocols such as TCP/IP, making the system vulnerable to hackers.

Strong passwords are not enough: In IoT networks, people use simple devices such as smart home appliances, which only include single-factor password support. People also do not pay attention to password support, and in most cases, owners do not prefer to change the default password, which makes them an easy target for attacks.

High performance, lightweight cryptography: IoT devices must perform for high-level data purposes, but due to resource constraint problems, they possess very lightweight cryptography.

Enabling secure updates: IoT devices have less memory and computation power, which does not allow them to receive constant updates, and owners may forget to update the devices. This allows attackers to install their own framework and update it according to their wishes. To date, several solutions have been proposed to secure the network. However, all of these have challenges, making it difficult for users to trust the IoT.

The main objective of the research was to develop a distributed IoT-based system by implementing the idea of blockchain for the security and privacy of the data resulting from the network of IoT. The motivation to carry out this research is that it can help to secure sensitive healthcare data, such as those produced during the COVID-19 pandemic. Implementing blockchain makes the data of IoT secure by making them unalterable and distributed.

This research work contributes by developing a prototype of a system that is distributed. The proposed system is decentralized and captures the data linearly. It not only stores the data in a time-stamped manner, showing the origin of the data, but also stores it immutable, preventing the modification of the entered record. Based on this system, auditing is easy because the entities involved can be held accountable. Making the system transparent provides openness to all devices.

## 2. Literature Review

The growth of IoT devices is continually increasing. According to Gartner [5], the number of new IoT devices rose from 1.691 billion in 2015 to 3.054 billion in 2019; the number of IoT devices exceeded the world population in 2020; the number of IoT devices exceeded the total number of PCs, tablet PCs, and smartphones in 2018, and reached 20.4 billion in 2020. The value of the IoT field is estimated to have reached USD 235 billion in 2021, with an annual growth rate of 16%. This third wave of the information technology industry will change how we see and observe the world based on accurate, immediate, and timely data. Moreover, in a recent report, Ericsson projected that, by 2024, 22.3 billion IoT devices would be connected to the Internet, with a compound annual growth rate (CAGR) of 17%. The greatest increases in cellular IoT devices, given in billions, are shown in Table 1.

**Table 1.** All IoT devices are connected to the Internet, as projected by Ericsson.

| IoT | 2018 | 2024 | CAGR |
|---|---|---|---|
| Wide-area IoT | 1.1 | 4.5 | 27% |
| Cellular IoT | 1.0 | 4.1 | 27% |
| Short-range IoT | 7.5 | 17.8 | 15% |
| Total | 8.6 | 22.3 | 17% |

In the COVID-19 pandemic situation, the number of IoT devices increased rapidly. Figure 1 shows the growth in IoT during the COVID-19 pandemic.

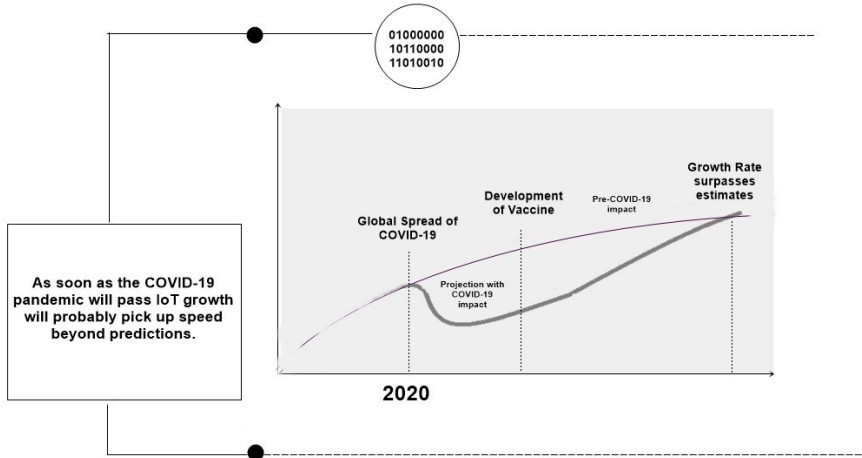

**Figure 1.** Growth in IoT in the COVID-19 pandemic [6].

IoT GPS can be used in ambulance systems, and automated robots can take care of COVID-19 patients. Blockchain is used to track medical supplies, facilitate increased testing and reporting, record the details of COVID-19 patients, enable a secure donation platform for supporters, limit supply chain disruptions, etc. [7]. Figure 2 describes the blockchain-based system for COVID-19 management [8].

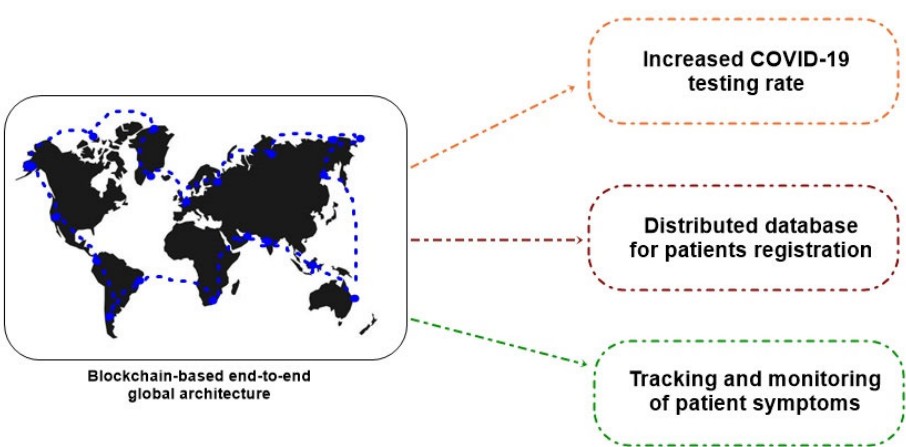

**Figure 2.** COVID-19 impact management using blockchain.

For a sensitive network such as IoT, it is necessary that the network is secured to provide bespoke services to the users. According to experts [9–11], data is the lifeblood of IoT. Thus, to physically secure the IoT heterogeneous devices, which do not have memory capacity and have almost no computation power, the idea is to safeguard the most important data. Roberta Rottigni stated that over 3000 people in Sweden selected a smart chip [10] to be embedded inside their bodies instead of using a card or several keys, giving us a clearer perspective of IoT than the field involving things. Now, this domain also includes the human being itself. IoT services will be a key technology for enabling the growth of smart cities [11], and will transform how we do business, sustain health, develop serious infrastructure, conduct education, and secure, defend, and amuse ourselves. Incorporating physical and cyber systems and human behaviors and interactions (e.g., producers, consumers, and attackers) will increase dependent infrastructure ecosystems' openness and attack surfaces. Building Automation Systems and Supervisory Control and Data Acquisition systems are the most common architectures to monitor and control intelligent structures such as smart homes and smart buildings. Building Automation Systems and Supervisory Control and Data Acquisition systems are interrelated with Internet resources and services, so they become easy targets for cyber adversaries. When discussing security

challenges, IoT faces the same cyber security challenges faced by traditional Internet users, including DoS attacks, malware, and jamming attacks [12]. IoT devices are also being used to launch DoS attacks [3] for other systems, which are also dangerous for IoT devices. Distributed denial of service attacks (DDoS) is a well-known form of attack on IoT devices. Sensors and RFIDs are primary devices for collecting data; thus, for DDoS attacks, it is elementary to perform jamming, kill commands, and de-synchronizing assaults [13]. Users' privacy is always a sensitive topic whenever there is a discussion about the challenges of IoT. Table 2 presents the existing security issues in IoT and their proposed solutions. Table 3 describes past and recent proposed IoT security solutions and their challenges.

**Table 2.** Security issues of IoT at different layers and their proposed solutions.

| Sr. No. | Security Problem | Result | Damage Layer | Proposed Solution |
|---|---|---|---|---|
| 1 | Jamming entities | Denial of service | Physical | Determining signal strength, calculating packet delivery ratio [14], encoding packets and change in frequencies [15]. |
| 2 | Insecure initialization | Denial of service and compromised the privacy | Physical | Establishing data transmission rates between nodes [16] and inserting fake noise [17]. |
| 3 | Sybil and spoofing attacks | Denial of service and network disturbance | Physical | Signal strength capacities [18], and channel evaluation [19]. |
| 4 | Insecurity of physical level | Denial of service and privacy compromised | Physical | Preventing software access to USB and preventing testing/debugging devices [20]. |
| 5 | Sleep deprivation | Energy expenditure | Link | Multi-layer interruption detection system [21]. |
| 6 | The duplicate attack caused by fragmentation | Denial of service and disturbance | Network | Insertion of timestamp for protecting against replay attacks [22], and fragment authentication through hash [23]. |
| 7 | Vulnerable neighbor discovery | IP spoofing | Network | Elliptic Curve Cryptography (ECC)-based signatures [24]. |
| 8 | Attack on RPL routing | Man-in-the-middle attack and monitoring | Network | Hashing with signature-built verification and observing node behavior [25]. |
| 9 | Wormhole and sinkhole outbreaks | Denial of service | Network | Rank verification through a hash function, trust level supervision, communication behavior evaluation of nodes, anomaly finding through IDS, and measuring signal strength [26–29]. |

**Table 2.** *Cont.*

| Sr. No. | Security Problem | Result | Damage Layer | Proposed Solution |
|---|---|---|---|---|
| 10 | Intermediate layer Sybil attack | Privacy breach, spamming, Byzantine errors, inaccurate broadcast | Network | Random social graphs, keeping lists of trusted/un-trusted users by analyzing user behavior [30,31]. |
| 11 | Authentic and secure communication | Privacy breach | Network | Compressed AH, IACAC using Elliptic Curve Cryptography, symmetric homomorphic mapping and distributed logs [32–34]. |
| 12 | Transport layer level security | Privacy compromised | Transport | DTLS-PSK with nonces, AES/SHA algorithm-based DTLS cipher, compressed IPSEC, and AES/CCM-based security [35–38]. |
| 13 | Session establishing and renewal | Denial of service | Transport | Verification with a prolonged secret key, and encryption-based symmetric key [39]. |

**Table 3.** Past and recent proposed IoT security solutions and their challenges.

| Techniques | Proposed Solutions | Challenges |
|---|---|---|
| Data Provenance | Provenance is a phenomenon in which the origin of the data along with the subsequent changes is traced in order to ensure the precision of data. Data provenance is about the creation of the propagation of the data process and where that data is serving, so this can be used in IoT to ensure confidentiality and integrity of data [40]. | A major challenge faced when implementing data provenance in IoT is that it is compulsory to also secure provenance data. An insecure data provenance means the exposure of sensitive data to unauthorized third parties [40]. |
| Blockchain | Using a distributed, decentralized, shared ledger that is accessible to all parties, blockchain will make it possible to share important relevant data collected from the IoT [41]. | Scalability and availability of blockchain [42]. |

**Table 3.** *Cont.*

| Techniques | Proposed Solutions | Challenges |
|---|---|---|
| Fog Computing | Fog computing's primary function is to locally manage the data produced by IoT devices for better administration, necessitating an architecture made up of various layers. The fog–device framework and the fog–cloud–device framework are two of their frameworks. The device and fog layer make up the former structure, whereas the device, fog, and cloud layer make up the latter framework. Layers are organized according to their capacity for storing and processing information. Wired or wireless communication is used for layer-to-layer communication [43]. | Fog computing inherits the security and privacy issues of cloud computing. The collection, transmission, processing, and exchange of users' sensitive data make privacy a crucial concern in fog computing. Owners of data are reluctant to reveal their privacy to outside parties, but privacy leakage is unheeded [43]. |
| Edge Computing | Edge computing, which places a small edge server between the user and the cloud or fog, is utilized as a solution to the issues with cloud computing. Instead of the cloud, some processing is undertaken at the edge server. The components of the edge computing architecture are edge devices, cloud servers, and fog nodes [44]. | Data security and user privacy are the two key issues with edge computing. The private information of a user may be exposed and used inappropriately if a home equipped with IoT devices is the target of cyberattacks [44]. |
| Machine Learning | In order to prevent data loss or other problems, the purpose of machine learning is to apply and train algorithms to detect anomalies in IoT devices or to detect any undesired activity taking place in IoT systems [45]. | Raw data of IoT devices cannot be processed by machine learning algorithms. Machine learning requires data to be classified and clustered. The network of IoT devices produces huge amounts of data and, before it is processed by machine learning algorithms, it needs to be cleaned and preprocessed accurately. Failing to do so will result in producing "garbage" data [45]. |

Previous research has found that IoT networks can be secured by applying data provenance, blockchain, fog computing, edge computing, and machine learning techniques. Table 3 presents the proposed solutions to overcome the IoT security issues.

The network of IoT produces a massive amount of data, which can be considered big data. For a large amount of data, such as big data, usual data processing techniques are not enough. Tas et al. [46] suggested a decentralized system and concluded that a centralized system would not help. Moreover, many frameworks working on new big data tools have been proposed to work with enormous amounts of data, such as supporting IoT, RIS-aided, and SLNR-based secure energy [47–50]. They claim that distributed tools will help with searching and querying the data.

## 3. Materials and Methods

This study used a decentralized solution to support IoT in different fields, especially healthcare. The essence of IoT lies in the data it collects. The system ensures the availability of the correct data so that decisions taken on the collected data are reliable. Research

assumes that data collection and decision-making are performed on the user side to decentralize the system, where services from a third party, such as the cloud, are not involved. The goal is to secure the IoT network, keeping in mind the resource-constrained behavior of IoT. A prototype using the programming language Python was developed to show the research results; this is an extra layer of security at the local store level. In blockchain technology, it is almost impossible to change the data accepted by all the nodes when they are not altered at the database level. Thus, it provides a basic level of data security in the system, ensuring that the data from resource-constrained devices can be trusted.

In the proposed model, all the devices are IoT devices with resource constraints, no computational power, low storage, and lower communication bandwidth. Suppose that the system is implemented in the traditional blockchain with complex algorithms, such as proof of work, requiring immense computational power; then, the idea of blockchain completely fails in the case of resource-constrained IoT devices. The proposed work and prototype system implements the basic concept of blockchain, which is decentralized, and the hash function, which is the backbone of blockchain. Each device that stores the data collected by the IoT device is a blockchain node; all of these storage devices make up the chain, and each new device is added at the end of the chain. When an IoT device collects data, that data string is sent and stored on every storage device, creating a decentralized system. When that string is entered and stored, a hash corresponding to that record is created and stored since the hash function ensures the same length of the line regardless of the size of the input string. Thus, it is almost impossible for attackers to guess or encrypt the original data. If the data are altered, the hash of that particular record is regenerated. This makes it impossible to modify or delete the data. The overall system works in three layers. The proposed IoT model is shown in Figure 3.

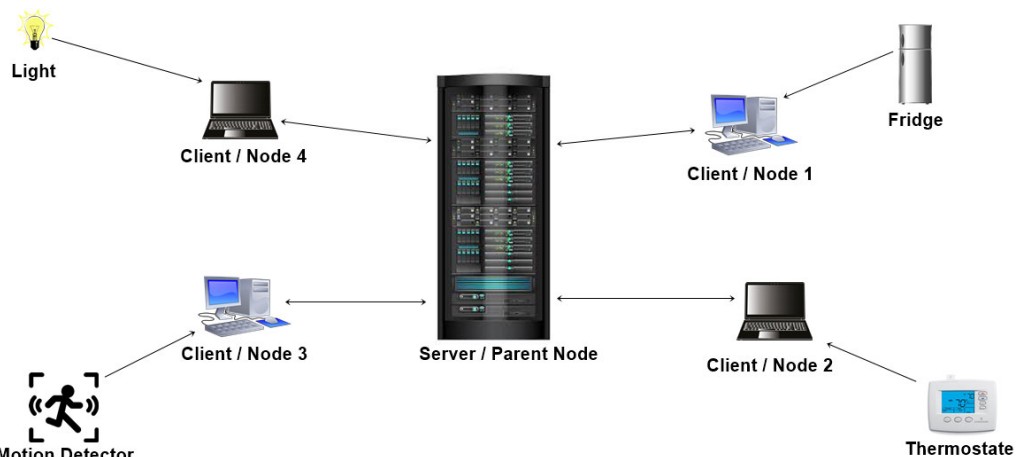

**Figure 3.** Proposed IoT model.

## 4. Implementation

This section explains the solution proposed in this study. Additionally, it is shown that data can be stored in a decentralized manner. The proposed solution is implemented in Python. The prototype is implemented using the package/library language support to implement a hash function and server-side implementation. The implemented code for the proposed system is broken down into three parts: client-side, server-side, and the code implemented to show the data entered.

### 4.1. Client-Side Implementation Code

The client-side code allows the client to connect to the chain. Client-side code implements socket programming. A socket is an endpoint in a communication network where two or more programs run on the same network. The socket on each side is bound to a specific port number; the port number enables the system to identify where to send those

data and which application is the destination. The client sends a request to the server for every connection, and further communication occurs if the server accepts the link.

*4.2. Server-Side Implementation Code*

The server hosts the request of the client. When the server is on, it listens to the new connection. When a client requests the link, the server connects to the clients, and communication occurs. Server-side codes were implemented through the socket library by the Python developers. The socket enables two-way communication by listening to the port destined for it. When a client requests the connection, the server listens to it and establishes a link that starts communication.

*4.3. GUI Implementation Code*

The third part of the code shows the graphical user interface. It helps all the nodes to view the data in the form of a table. Here, the GUI shows the results of the connected systems. It offers the three related systems, named Shayan, Ayesha, and Arslan, including manually added Device Names and the Key, and automatically showing the Name and the Time the data was entered. Figures 4–8 illustrate the working of the proposed blockchain method applied to IoT. Every record has a hash responding to it.

A new connection joins, named Faizan, showing the timestamps and data entered. These data can be seen through every other node connected to the chain. In the table showing data, the entities "Time" and "Entered From" can never be changed by the user nodes. The entities "Key" and "Device Name" are the data entered and stored. Each record shown in the table has a hash value that is stored and visible only to the parent node. The following figure shows the corresponding hash values of each record entered.

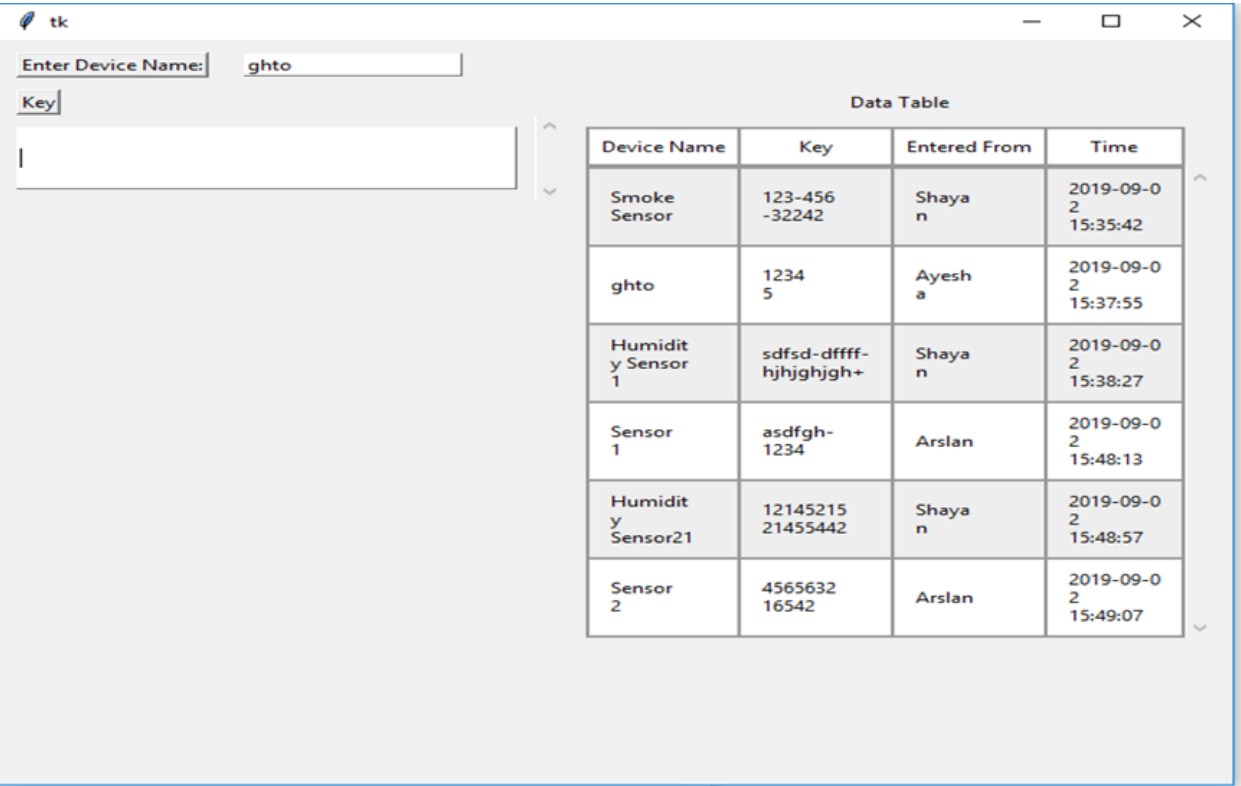

**Figure 4.** Data of the chain.

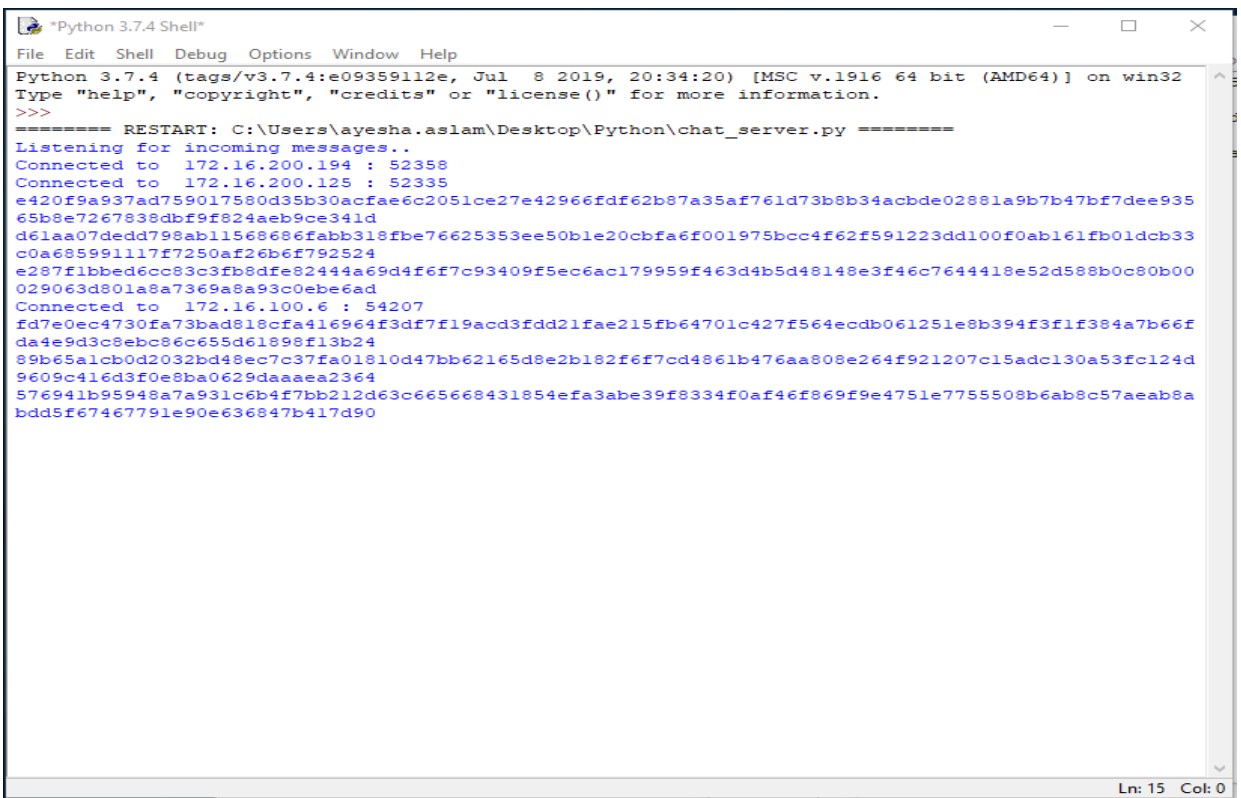

**Figure 5.** Hash for every record.

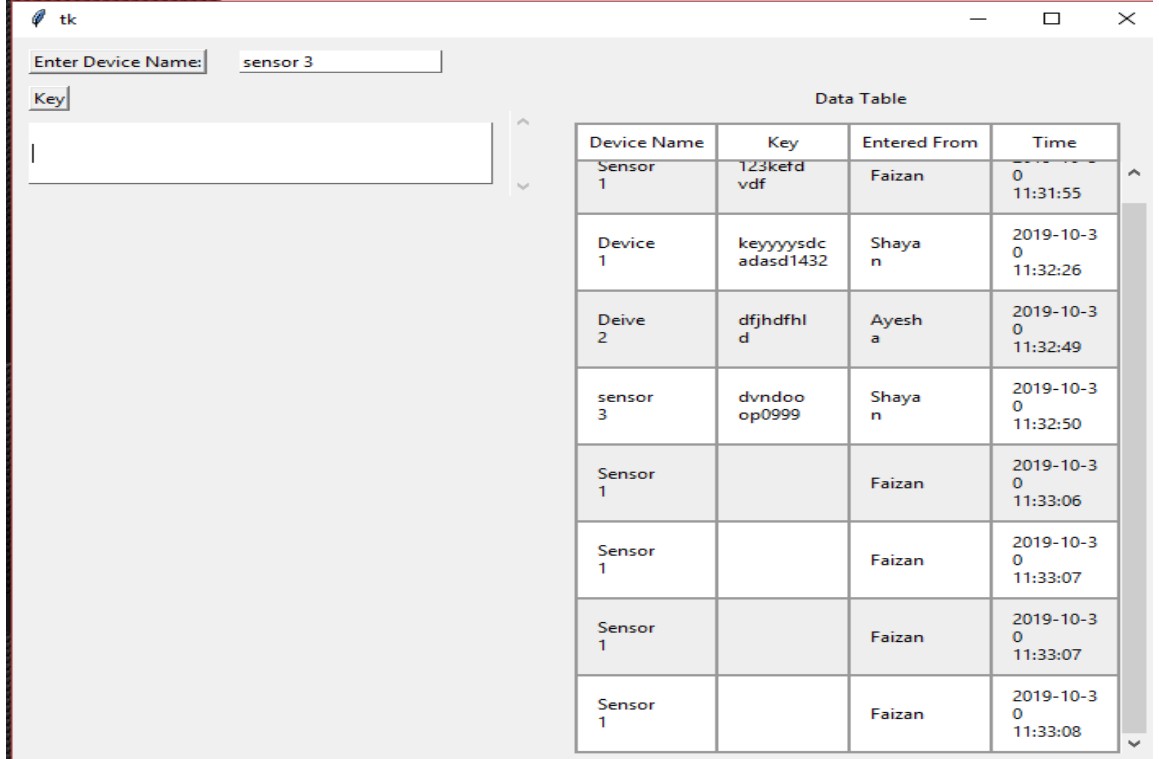

**Figure 6.** New node joined.

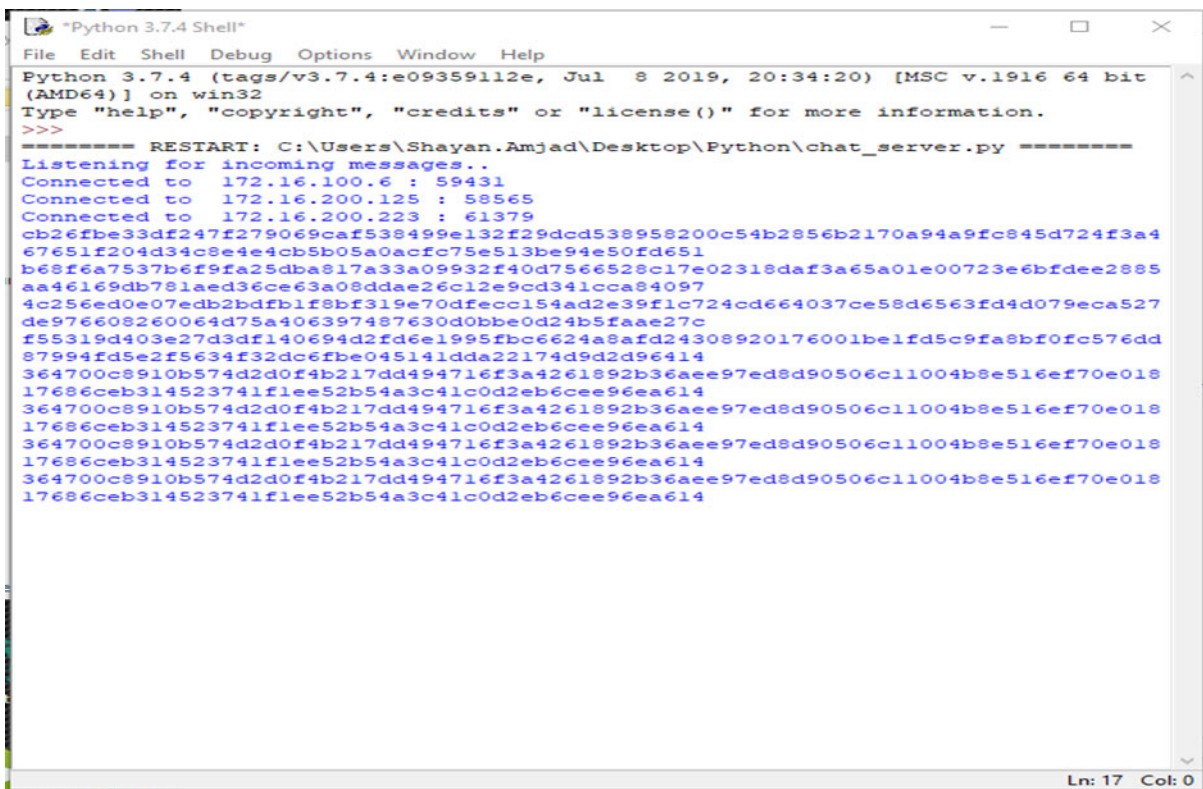

**Figure 7.** Hash of record entered by the new node.

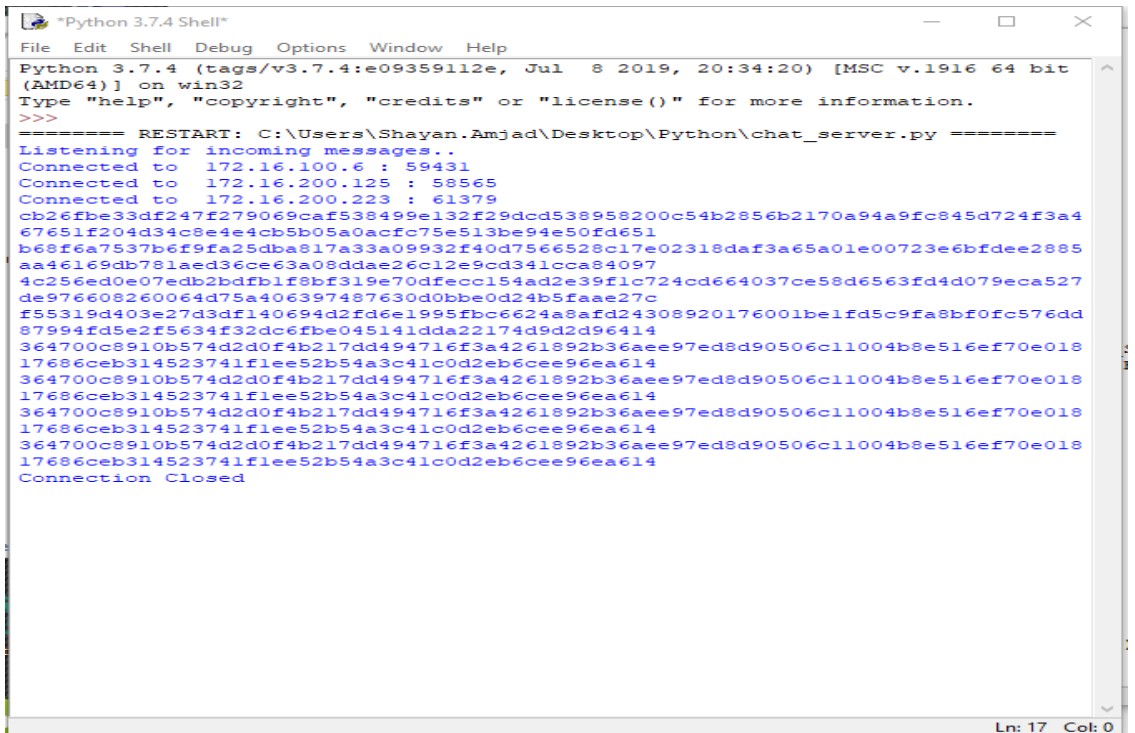

**Figure 8.** One node disconnected.

When the new node joins and enters data, every activity is kept in the parent node, and every hash value is saved. This is shown in Figure 8 even if the node is disconnected and no longer connected to the system. The storage device still has the data entered by the node and their corresponding hash values.

When the node is disconnected server-side, or the parent node shows the message "Connection Closed", entry of the data by the node is prevented. This is shown in Algorithm 1.

---

**Algorithm 1:** Entry of the data

---

```
                   Server
While(1)
      i = 0, data = null
      createsocket()
      communicateSocketAddress()
      listeningIncommingConnection()
      ConnectClient()
while(i < no. of client)
{
      Data = listen (client [i])
            If data ≠ Ø
            Generate hash()
            J = 0
                  While(i < no. of clients)
                        If(i ≠ j)
                        Share(data, client[i])
            Exit()
}
Client
While(1)
{
      createClientSocket()
      ConnectServer(IP, port)
      Save(data)
      Share(data)
}
closeConnection()
```

---

A step-by-step explanation of the algorithm:

Server

1.  Create a socket
2.  Share the socket address and continue to watch for incoming connection requests
3.  Link to client
4.  Data received
5.  Create a hash for each received string
6.  Share the received data with every other connected node
7.  Repeat steps 5 and 6 as desired by the user.
8.  Exit.

Client/Nodes

1.  Make a distinct client socket for each node.
2.  Using the provided socket address, connect to the server (IP and port)
3.  Data sent and received.
4.  Repeat step 3 as configured.
5.  Close connection.

## 5. Analysis and Discussion

For an ever-growing technology such as IoT to succeed, it is necessary to make it secure. To date, all research that proposed implementing the blockchain for the network of IoT used a consensus algorithm to validate new nodes. The consensus algorithm requires computational power, and the addition of new nodes results in the non-availability and scalability of the chain. This research work presents a solution to solve the problem of

security in IoT by making the system decentralized without implementing a consensus algorithm. The prototype implements a more straightforward cryptographic function known as a hash function. The hash function ensures the data's security by mapping a variable-length string to a string having a fixed length. This mechanism makes the decryption hard for attackers, and it is almost impossible to guess the actual line. The prototype shows that, with decentralized technology, all the data will be linearly appended at the end of the chain, and cannot be changed once entered. Moreover, the data cannot be deleted once entered, provided it is kept on-chain. As a result, the data of the system can be used reliably.

In the case of damage to one system, the data is safe on the other points. In blockchain technology, each transaction has a saved hash value. The proposed solution applies an extra security layer on the storage side. All the storage devices that collect and save IoT data have a decentralized system. All these storage devices are connected when an IoT device enters data held in any storage device; these data will be updated on each storage device attached. On the same side, each record entered by the machines will create a hash value. This hash value will be stored in the parent node/server. The hash value is never the same as the actual data, so no one can access or alter the data. When information is deleted or changed on the storage device, the hash value will remain on the parent node, and decrypted to obtain the original data. Even if the system disconnects the devices, these data are saved and available. The data collected by the devices are stored in raw form, which can be used to obtain insights into the data and maximum results.

## 6. Conclusions

During the COVID-19 pandemic, it was difficult for people to physically perform most work. As a result, most people moved their work online. IoT currently plays an important role, and many people benefit from it. A large amount of data is converted from physical to online. As the data rate increases, the data need to be managed and secured more efficiently. Blockchain plays an essential role, in this context, in ensuring people's data on the Internet. A literature review shows that many techniques have been proposed to secure the network of IoT, each of which has shortcomings, meaning the network of IoT is not reliable. As a tamper-proof distributed and shared database, blockchain has been proven to be the best fit for IoT due to its benefits. Blockchain provides necessary features for security, including transparency, immutability, and audibility. Data stored in a decentralized manner do not allow modification of the records and maintain the history, which fulfills the concept of provenance.

Hence, the proposed study designed a system to use blockchain to make IoT data reliable. As a result, the network of IoT is an acceptable solution, especially in healthcare (COVID-19) and for people who have moved their work online. This study contributes to the security of data produced by IoT devices, but it still has many shortcomings which can be researched and addressed. The approach used in this study can be authenticated by professionals using the system of IoT. The organizations implementing the IoT system to collect data and gain insights into the data will further elaborate on the system's shortcomings.

**Author Contributions:** Conceptualization, H.A.M.M.; Methodology, A.A.S.; Software, A.K.; Formal analysis, A.A.S., A.M. and A.K.; Resources, A.A.; Data curation, A.A.; Writing—original draft, A.A.S. and A.A.; Writing—review & editing, H.A.M.M., A.M. and A.K.; Project administration, H.A.M.M. and A.M. All authors have read and agreed to the published version of the manuscript.

**Funding:** This research received no external funding.

**Data Availability Statement:** Not applicable.

**Conflicts of Interest:** The authors declare no conflict of interest.

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
