# Peer review of "Resolving Security Issues in the IoT Using Blockchain"

_electronics, doi:10.3390/electronics11233950_

Round 1

Author Response

We are thankful for reviewing our manuscript. We greatly appreciate the comments and suggestions you have provided. The manuscript has been updated accordingly. Thank you for your consideration.

Reviewer 2 Report

In the literature review, past and recent related works must be presented in a tabular format as a summary with different heads in the table.

Highlight points-wise the existing security issues in IOT without using blockchain clearly as a result of the literature survey.

Fig 1. is the copy pasted diagram from "https://www.weforum.org/agenda/2021/03/what-is-the-internet-of-things/".

similarly, check fig2 and fig3 for the same issue as in fig1.

Fig 4. "IoT-based System Architecture" is the exact copy of the figure given in the published paper "DOI: 10.3390/en11051252". 
This turns out to be a violation of copyright if the author has not taken written permission.

The author should redraw the diagram with their own inputs and can be represented.

Figure 5. "Three layers of the model"  needs to be redrawn to show wing adding components at granular layer clarity to the clarity to audience about the contribution.

Author Response

(The authors gave the same response as above.)

Reviewer 3 Report

The drafting and organization of the work are more generic and not up to the standard.

It is advised authors add the algorithm of steps of code (Python) and explain each and every step of the implementation process.

There is no comparison table of proposed work with recent similar works, that can add value to the proposed work.

Reduce the general information about IoT and Block Chain and enhance the quality of work by adding what exactly this work speaks to make it more interesting to the readers.

Figure captions could be " Figure 1, 2, 3... so on.. but not Figure 6.1, 6.2.....etc..

Need more explanation about the figures and they are also not clear...

Author Response

(The authors gave the same response as above.)

Round 2

Reviewer 1 Report

The authors have well addressed my concerns, no further comments.

Author Response

The response2 file is attached.

Reviewer 3 Report

The authors responded to the all the comments, but figures are not clear. Can be improved with higher resolution. 

Author Response

The response2 file is attached.
